# New Type of Tannins Identified from the Seeds of *Cornus officinalis* Sieb. et Zucc. by HPLC-ESI-MS/MS

**DOI:** 10.3390/molecules28052027

**Published:** 2023-02-21

**Authors:** Jun Li, Lin Chen, Hua Jiang, Min Li, Lu Wang, Jia-Xing Li, Yue-Yue Wang, Qing-Xia Guo

**Affiliations:** Department of Pharmaceutical Engineering, College of Chemistry and Chemical Engineering, Henan University of Science and Technology, Luoyang 471023, China

**Keywords:** seed extract, *Cornus officinalis* Sieb. et Zucc., polyphenol, mass spectrometry, tannin

## Abstract

There is a lack of information on the compound profile of *Cornus officinalis* Sieb. et Zucc. seeds. This greatly affects their optimal utilization. In our preliminary study, we found that the extract of the seeds displayed a strong positive reaction to the FeCl_3_ solution, indicating the presence of polyphenols. However, to date, only nine polyphenols have been isolated. In this study, HPLC-ESI-MS/MS was employed to fully reveal the polyphenol profile of the seed extracts. A total of 90 polyphenols were identified. They were classified into nine brevifolincarboxyl tannins and their derivatives, 34 ellagitannins, 21 gallotannins, and 26 phenolic acids and their derivatives. Most of these were first identified from the seeds of *C. officinalis*. More importantly, five new types of tannins were reported for the first time: brevifolincarboxyl-trigalloyl-hexoside, digalloyl-dehydrohexahydroxydiphenoyl (DHHDP)-hexdside, galloyl-DHHDP-hexoside, DHHDP-hexahydroxydiphenoyl(HHDP)-galloyl-gluconic acid, and peroxide product of DHHDP-trigalloylhexoside. Moreover, the total phenolic content was as high as 79,157 ± 563 mg gallic acid equivalent per 100 g in the seeds extract. The results of this study not only enrich the structure database of tannins, but also provide invaluable aid to its further utilization in industries.

## 1. Introduction

Studies have shown that plant-based diets rich in polyphenols can exert health-promoting effects by reducing the risk of many diseases, such as cancer and neurodegenerative, cardiovascular, and inflammatory diseases. Therefore, it is vital to explore new sources of bioactive plant polyphenols and carry out their characterization for promoting human health [1,2,3].

*Cornus officinalis*, also known as Asiatic dogwood, is a deciduous shrub in the genus *Cornus* of the family Cornaceae that is mainly distributed in China, Korea, and Japan [4]. The pericarp of its fruit is used as a traditional Chinese herbal medicine that is widely used clinically along with other herbal medicine to treat different symptoms. For example, it has been used in combination with Mantidis Oötheca, Rubi Fructus, and Rosae laevigatae Fructus to clinically treat the urinary bladder dysfunction. It is prescribed together with Radix Rehmanniae Praeparata, Dioscoreae Rhizoma, Alismatis rhizoma, Moutan Cortex, and Poria to treat patients with vertigo, tinnitus, and waist and knees weakness [5]. Because of its wide traditional clinical use, many phytochemical and pharmacological studies have been conducted on the fruit pericarp. To date, about 90 compounds have been isolated and identified and are classified as terpenoids, flavonoids, tannins, polysaccharides, phenylpropanoids, sterols, and carboxylic acids, with iridoids, tannins, and flavonoids being the major compounds [6]. They display a wide range of pharmacological activities, such as hypoglycemic [7], antibacterial [8], hypolipidemic [9], antioxidant [10], anticancer [11], neuroprotective [12], and hepatoprotective activities [13].

In contrast, few reports have been published on the seeds because of their minor applications. The seeds account for approximately 50% of the fresh fruit’s weight. It is estimated that approximately 6000 tons of seeds are generated annually, owning to the huge pericarp consumption in the Chinese medicine industry [14]. Moreover, seed use meets the 12th sustainable development goal (SDG 12), sustainable consumption and production of the United Nations (UN) 2030 Agenda for Sustainable Development 2015 [15]. Therefore, there is an urgent need to develop optimal processing methods for the valorization of seeds.

Notably, fruit seeds can be converted into biofuels. Production of bio-oil from the seeds of cherry plum and peach has been reported [16,17,18,19]. However, based on our observations of the seeds of *C. officinalis*, we found that each seed has a tiny kernel surrounded by a thick wall of the lignified endocarp. Biofuel conversion of the seeds is not feasible, owing to the tiny kernel. The kernel is the main source for biofuel conversion, as it comprises abundant fatty acids; however, it accounts for less than 5% of the seed weight. Nevertheless, we found a water-soluble yellow powder substance in the cavities of the thick endocarp, which accounts for 40% of the endocarp weight. Moreover, it gives a strong positive reaction to FeCl_3_ solution (6C_6_H_6_OH + FeCl_3_ → H_3_[Fe(C_6_H_6_O)_6_] (purple color) + 3HCl), indicating the presence of polyphenol.

However, only nine polyphenols have been reported from the seeds of *C. officinalis*: 1,2,3-tri-*O*-galloyl-*β*-*D*-glucose, 1,2,6-tri-*O*-galloyl-*β*-*D*-glucose, 1,2,3,6-tetra-*O*-galloyl-*β*-*D*-glucose, 1,2,4,6-tetra-*O*-galloyl-*β*-*D*-glucose, 1,2,3,4,6-penta-*O*-galloyl-*β*-*D*-glucose, tellimagrandin II, gallic acid 4-*O*-*β*-*D*-glucoside, and gallic acid 4-*O*-*β*-*D*-(6′-*O*-galloyl)-glucoside [20]. Moreover, ellagic acid can be detected abundantly in its acid-hydrolyzed sample, which indicates that it is rich in ellagitannins [21].

In our pre-experimental study, we found that the number of reported polyphenols was far less than the number of polyphenols detected by HPLC in the pre-experiment. Therefore, the main objectives of the present study were to characterize and identify polyphenols in the seeds of *C. officinalis* and to provide valuable information for its use on an industrial scale, such as antioxidant additives in food or drugs.

The HPLC-ESI-MS/MS is a powerful tool used for the separation and identification of polyphenols in plant extracts and can provide an invaluable contribution to polyphenol analysis. It was employed as the main investigation tool to achieve the study objectives [22].

## 2. Results and Discussion

### 2.1. General

In this study, a total of 90 phenolic components were identified using coupled chromatographic and mass spectrometric analysis of the water-soluble extract obtained from the seeds of *C. officinalis*. They were classified into nine brevifolincarboxyl tannins and their derivatives, as well as 34 ellagitannins, 21 gallotannins, and 26 phenolic acids and their derivatives. Among them, we reported five new types of tannin for the first time: brevifolincarboxyl-trigalloyl-hexoside, digalloyl-dehydrohexahydroxydiphenoyl (DHHDP)-hexdside, galloyl-DHHDP-hexoside, DHHDP-hexahydroxydiphenoyl(HHDP)-galloyl-gluconic acid, and the peroxide product of DHHDP-trigalloylhexoside.

The polyphenols were identified based on their chromatographic profiles, their MS data of [M-H]^−^, and their MS/MS fragmentation profiles by comparing with published data. Notably, the obtained deprotonated polyphenol molecules and their typical cleavage of precursor ions accelerated their identification. The MS spectrum of the brevifolincarboxyl moiety was first revealed by the specific fragment ions at *m*/*z* 247, 273, and 291. The DHHDP moiety in the tannin structure was indicated by the fragment ions of brevifolincarboxyl moiety, together with a fragment ion, indicating a 44-Da mass loss from the [M-H]^−^ resulting from rearrangement and decarboxylation, and this greatly helped in the identification of the DHHDP moiety. The fragment ions at *m*/*z* 249.03, 275.02, and 300.99 are typical of the HHDP moiety. The galloyl moiety was revealed by the fragment ions at *m*/*z* 169.01 and 125.02. Furthermore, the number of galloyl structures in tannin can be determined by a group fragment ion representing a 152-Da mass difference, indicating that consecutive galloyl moieties are lost. The 44-Da mass loss from the pseudo-molecular ion is characteristic of phenolic acid.

The total ion chromatogram of the seed water extract of *C. officinalis* in the negative ESI model is illustrated in Figure 1. Compound identification within each class is detailed below and summarized in Table 1.

### 2.2. Brevifolincarboxylic Tannins and Their Derivatives

An [M-H]^−^ ion at *m*/*z* 909.1014 with a retention time of 19.62 min was observed for **A1-1**, producing fragment ions at *m*/*z* 757.0958, 604.7536 and 453.0302, indicating the consecutive loss of three galloyl moieties (152 Da) from the [M-H]^−^ ion [3]. Additionally, typical fragment ions for brevifolincarboxyl moiety (274 Da) at *m*/*z* 247.0257, 273.0030, and 291.0139 were exhibited (see **D8**) [23]. Additionally, a hexose core (180 Da) can be determined based on the mass difference between the molecular weight (910 Da) and the total weight of the determined moieties (730 Da). Therefore, **A1-1** was putatively assigned as a brevifolincarboxyl-trigalloyl-hexoside. Moreover, we found a fragment ion at *m*/*z* 435.0560 that resulted from the loss of H_2_O from the fragment ion at *m*/*z* 453.0302. This shows the presence of brevifolincarboxyl-hexoside moiety, hence supporting the proposed structure. Furthermore, two other compounds **A1-2** and **A1-3**, with retention times of 21.28 min and 23.80 min, respectively, displayed the same pseudomolecular fragment ion and fragment patterns, indicating the occurrence of two brevifolincarboxyl-trigalloyl-hexoside isomers. Regarding the structures of the three isomers, the differences were based on the position of the linkages of the three gallic and brevifolincarboxyl moiety to the hexose core.

To the best of our knowledge, brevifolincarboxyl-trigalloyl-hexoside-type tannins have not been reported. The only two analog compounds reported are decarboxylated geraniin, a product of geraniin treated with sodium benzenesulfinate [24], and repandusinin from the genus *Mallotus* [25].

Hydrolysable tannins include gallotannin (GT) and ellagitannin (ET). They are the polyol esters, usually of glucose or quinic acid [26], with the moieties of HHDP and gallic acid [22]. **A1** features a tannin with a brevifolincarboxyl moiety linked to hexose, which has not been widely described. The MS/MS data of **A1-1** revealed that the fragment ions at *m*/*z* 247, 273, and 291 could be used as typical indicator ions for identifying a brevifolincarboxyl moiety in a tannin structure [3,27]. The **A1-1** MS fragment pattern is shown in Figure 2.

**A2** exhibited an [M-H]^−^ ion at *m*/*z* 605.0013, with a retention time of 18.36 min, that released a fragment ion at *m*/*z* 331.0200, corresponding to a monogalloyl-hexoside moiety (**C1-1**). This resulted from the loss of a brevifolincarboxyl moiety (274 Da) from the [M-H]^−^ ion. Monogalloyl-brevifolincarboxyl-hexoside was tentatively assigned to **A2**. Moreover, a fragment ion at *m*/*z* 453.0401 attributed to the brevifolincarboxyl-hexoside moiety, resulting from the loss of a galloyl unit from its pseudo-molecular ion. This supported the identification of **A2**. Typical fragment ions for galloyl moiety were found at *m*/*z* 169.0128 and 125.0230. Those for the brevifolincarboxyl moiety were observed at *m*/*z* 247.0255, 273.0026, and 291.0116 and supported the identification of **A2**. The fragment ions at *m*/*z* 587.0662 and 435.0555 were obtained from the consecutive loss of H_2_O (18 Da) and galloyl (152 Da) moieties, respectively, from the pseudomolecular ion. Monogalloyl-brevifolincarboxyl-hexoside, a tannin type with the structure of 1-*O*-galloyl-4-*O*-brevifolincarboxyl-*β*-*D*-glucoside, has previously been isolated and characterized in the leaves of *Marcaranga tanarious* (L.) MUELL *et* ARG. [28]. However, to the best of our knowledge, this type of tannin has not been reported in the seeds of *Cornus officinalis* Sieb. et Zucc.

An [M-H]^−^ ion at *m*/*z* 801.0795 with a retention time of 17.01 min was observed with **A3**. The fragment ions produced at *m*/*z* 247.0251, 273.0023, and 291.0128 indicated the presence of a brevifolincarboxyl moiety, similar to **A1-1** and **A2**. At the fragment ion 435.0555, dehydrated brevifolincarboxyl-hexoside was revealed, as mentioned in **A2**. This was speculated to be formed by the loss of two consecutive galloyl moieties and H_2_O from the fragment ion at 757.0929. However, the mass difference of 44 Da indicated that the fragment ion at 757.0929 was formed by the loss of a carboxylic moiety from the pseudo-molecular ion at *m*/*z* 801.0795. Thus, it can be inferred that the brevifolincarboxyl group did not constitute the final structure of **A3**. Based on the reaction of geraniin with sodium benzenesulfinate [24], we suggested that a brevifolincarboxyl group was formed from DHHDP by rearrangement and decarboxylation (Figure 3). Therefore, **A3** was tentatively identified as digalloyl-DHHDP-hexoside.

Based on the above, a simple and reliable method to identify a DHHDP moiety in tannin is characterized by typical fragment ions at *m*/*z* 247, 273, and 291 for the brevifolincarboxyl moiety and a 44-Da mass difference between the [M-H]^−^ ion and a decarboxylated fragment ion based on **A3** identification. **A3** with a tannin-type digalloyl-DHHDP-hexoside has not been previously reported, to the best of our knowledge.

**A4** displayed a pseudo-molecular ion at *m*/*z* 953.0915, with a retention time of 19.62 min, that produced the typical fragment ions for brevifolincarboxyl moiety at *m*/*z* 247.0257, 273.0031, and 291.0140. Moreover, a fragment ion at *m*/*z* 909.0996, 44-Da mass lower than its [M-H]^−^, was observed, indicating the presence of a DHHDP moiety. Fragment ions at *m*/*z* 757.0971, 605.0547, and 435.0559 resulted from the sequential loss of three galloyl moieties from the fragment ion at 909.0996, corresponding to the decarboxylated [M-H]^−^ ion. Moreover, a fragment ion at 435.0559 was also observed, which is typical of the dehydrated brevifolincarboxyl-hexoside observed in **A1-1**, **A1-2**, **A1-3**, **A2**, and **A3**. Therefore, **A4** was tentatively identified as a DHHDP-trigalloylhexoside. This study is the first to report this finding from the seeds of *C. officinalis*. The only type of tannin in DHHDP-trigalloylhexoside is isoterchebin with a structure of 1,2,3-*O*-galloyl-4,6-*O*-DHHDP-*β*-*D*-glucose, which has been reported in the fruit of *Cornus officinalis* Sieb. et Zucc. by Okuda, 1981 [29].

The chromatogram and MS/MS profile of **A5** showed an [M-H]^−^ ion at *m*/*z* 649.1073, with a retention time of 18.36 min. It was tentatively identified as a monogalloyl-DHHDP-hexoside based on the fragment ions at *m*/*z* 497.0930, corresponding to DHHDP-hexoside. This fragment resulted from the loss of a galloyl moiety from its pseudo-molecular ion. The fragment ions at *m*/*z* 169.0128 and 125.0230 were typical of galloyl moiety. To our knowledge, the tannin-type monogalloyl-DHHDP-hexoside has not yet been reported.

**A6** exhibited an [M-H]^−^ ion at *m*/*z* 967.1068 with a retention time of 22.55 min. The fragment ions at *m*/*z* 249.0377, 275.0203, and 300.9964 resulted from the ellagic acid moiety, indicating the occurrence of the HHDP moiety in **A6**. The mass differences of 152 and 326 Da (=318 Da + 18 Da) between two pairs of fragment ions at *m*/*z* 765.0391, 917.0799 581.0547, and 917.0799, respectively, indicating the loss of a galloyl moiety and a DHHDP moiety. Based on these results, the mass difference of 196 Da between its pseudo-molecular weight (967 Da) and the total weight (771 Da) of the identified moieties of HHDP, galloyl, and DHHDP indicated the presence of a gluconic acid moiety [30]. Therefore, **A6** was tentatively assigned as a DHHDP-HHDP-galloyl-gluconic acid. The group fragment ions at *m*/*z* 247.0253, 273.0068, and 291.0096, corresponding to the brevifolincarboxyl moiety, confirmed the occurrence of the DHHDP moiety.

Tannin-type DHHDP-HHDP-galloyl-gluconic acid of **A6** has not been reported to date. A typical mass loss of 318 Da was observed with the DHHDP moiety [3]. This tannin type is characterized by a gluconic acid as the polyol core, that is rarely reported in the tannin structure. Lagerstannin C (galloyl-HHDP-gluconic acid) from *Lagerstroemia speciosa* L. pers, punigluconin (digalloyl-HHDP-gluconic acid) from pomegranate (*Punica granatum* L.) peel, and 12 mixed HHDP-galloylgluconic acids from the jabuticaba species are examples of gluconic acid as the core of tannin [31].

As indicated by the MS/MS spectrum of **A7**, it had a retention time of 23.39 min and comprised almost all the fragment ions that were liberated by **A1**, such as ions at *m*/*z* 909.1002, 757.0970, 605.0627, 435.0560, 291.0138, and 247.0257. Therefore, it can be inferred that **A1** and **A7** were structurally similar. The only difference between **A1** and **A7** is the [M-H]^−^ ion. **A7** exhibited an [M-H]^−^ ion at *m*/*z* 941.1275, while that of **A4** was 12 Da lower. Based on the fragmentation pattern of **A4**, it was assumed that **A7** was a peroxide product of DHHDP-trigalloylhexoside (Figure 4), which should be regarded as an intermediate product in the decarboxylation process from **A4** to **A1**.

Possible structures of the tannin-type **A1**–**A7** are illustrated in Figure 5. Regarding the polyphenol structure of tannin, the moieties attached to the polyol are galloyl group in gallotannin (type I), HHDP group in ellagitannin (type II), DHHDP group in dehydroellagitannin (type III), and transformed DHHDP group in transformed dehydroellagitannin (type IV) [32]. Herein, we first report the occurrence of brevifolincarboxyl moiety as the substituent to the hexose core in **A1-1**, **A1-2**, **A1-3**, and **A2** from the seeds of *C. officinalis*. To date, there are only two reported brevifolincarboxyl tannin: 1-*O*-galloyl-3,6-HHDP-4-*O*-brevifolincarboxyl-*β*-*D*-glucopyranose, that is the basic hydrolytic product of geraniin and repandusinin from the genus *Mallotus* [24,25]. Our study promoted the brevifolincarboxyl tannin structure diversity, which can be classified as a new type V tannin. Based on the DHHDP moiety fragment pattern in **A3**, **A4**, and **A5**, the brevifolincarboxyl moiety is thought to be biosynthetically derived from the DHHDP moiety by rearrangement decarboxylation and lactonization [24]. The bio-relationship can be then illustrated in Figure 6.

### 2.3. Ellagitannins

**B1-1** showed [M-2H]^2–^ at an *m*/*z* of 708.0711 with a retention time of 9.6 min, corresponding to a molecular weight of 1418 Da. The produced mono-charged fragment ion at *m*/*z* of 785.0749 corresponded to a valoneoyl-galloyl-hexoside, such as isorugosin B [33], without resulting from the loss of the HHDP-galloyl-hexose moiety (e.g., gemin D, **B2**). This enabled the tentative identification of **B1-1** as a dimer composed of a valoneoyl-galloyl-hexoside and HHDP-galloyl-hexoside, such as camptothin A [34]. The fragment ions at *m*/*z* 450.9911, 300.9971, and 633.078 indicated the presence of valoneoic acid trilactone (VTL), HHDP, and HHDP-galloyl-hexoside moieties, respectively, which supported the proposed identification of **B1-1**. **B1-2** and **B1-3** had the same [M-2H]^2–^ at *m*/*z* 708.07 and showed similar fragmentation patterns at retention times of 10.06 and 10.69 min, respectively. These indicated the other two isomers of **B1-1**.

Furthermore, the HHDP moiety was observed to be part of **B2-1** and **B2-2**, as they both showed an [M-H]^−^ ion at *m*/*z* 633 and fragment ions at *m*/*z* 300.99, 275.01, 249.03, 169.01, and 125.02. These were typical of the HHDP and gallic acid moieties. The fragment ion at m/e 331.06 was attributed to monogalloyl-hexoside, due to the loss of an HHDP moiety from the [M-H]^−^ ion. Thus, **B2-1** and **B2-2** were identified as isomers of HHDP-monogalloyl-hexoside-type tannins, such as gemin D [35]. The fragment ion was observed at *m*/*z* 481.05, corresponding to HHDP-hexoside, resulting from the loss of a galloyl moiety (152 Da), thus supporting their identification. **B2-1** and **B2-2** differed in the linkage of the HHDP and galloyl moieties to the hexoside core.

**B3** had an [M-H]^−^ ion at *m*/*z* 783.0750 with a retention time of 10.09 min. The fragment ions at *m*/*z* 481.0508 corresponded to the HHDP-hexoside moiety. It was the result from the loss of one HHDP moiety from the pseudo-parent ion. **B3** was then tentatively identified as a bis-HHDP-hexose-type tannin [22]. The fragment ion at *m*/*z* 300.9970 was typical for ellagic acid, indicating the occurrence of the HHDP moiety. Dissociation of the ion at *m*/*z* 300.9970 yielded an *m*/*z* 257.0208 (loss of 44 Da, free carboxyl unit), which is characteristic of LHHDP produced by the loss of 44 Da, a free carboxyl unit, from ellagic acid.

**B4-3** with a retention time of 16.19 min was characterized as a digalloyl-HHDP-hexoside-type tannin, as with tellimagrandin I as an exemple [36]. This identification was possible based on its [M-H]^−^ ion at *m*/*z* 785.0776 and the release of typical fragment ions at *m*/*z* 483.1278 corresponding to an HHDP-hexoside moiety, resulting from the loss of two galloyl moieties (152 Da) from the [M-H]^−^ ion. Typical fragment ions at *m*/*z* 300.9963, 275.0201, and 249.0407 confirmed the appearance of the HHDP moiety in **B4-3**. **B4-3** has other two isomers, **B4-2** and **B4-1**, with retention times at 11.65 and 13.85 min, respectively, that showed a similar fragment pattern as **B4-3**.

The molecular weight of **B5-1** was determined to be 1570 Da based on the doubly deprotonated ion at *m*/*z* 784.0739 with a retention time of 12.51 min. The fragment ion at *m*/*z* 785.0741 corresponded to an HHDP-digalloyl-hexoside moiety, such as tellimagrandin I (**B4**), indicating that **B5-1** was a dimer composed of two HHDP-digalloyl-hexoside moieties by the elimination of H_2_. The fragment ion at *m*/*z* 450.990, attributed to valoneic acid tridilactone [VTL-1]^−^, indicated the occurrence of a valoneoyl bridge. Thus, **B5-1** was tentatively determined as an HHDP-digalloyl-hexoside dimer type tannin, such as cornusiin A [34]. Moreover, the fragment ions at *m*/*z* 633.06 and 300.9970 attributed to an HHDP-digalloyl-hexoside moiety, such as gemin D (B2), and ellagic acid further supported the proposed **B5-1** structure. Additionally, there were five other isomers or anomers of **B5-1**, with retention times of 13.46, 14.38, 15.51, 17.95, and 19.62 min, that showed identical MW and fragmentation patterns.

**B6** displayed a molecular weight of 1086 Da, based on the doubly deprotonated ion at *m*/*z* 542.03, with a retention time of 14.38 min. The fragment ion at *m*/*z* 785.0763 was attributed to the HHDP-digalloyl-hexoside moiety, such as tellimagrandin I (**B4**), which resulted from the loss of the EA moiety from the pseudo-parent ion. The cornusiin B isomer was tentatively assigned as **E13** [37]. The fragment ions at *m*/*z* 633.0634, 450.970, and 300.997 indicated the appearance of gemin D, VTL, and EA moieties, respectively, which supported the proposed structure.

**B7** exhibited a pseudomolecular ion at *m*/*z* 953.0909. The fragment ion *m*/*z* at 785.0754 resulted from the loss of a valoneoyl moiety, which was supported by the fragment ions at *m*/*z* 909.0972, corresponding to the loss of 44-Da carboxyl unit. The fragment ions at *m*/*z* 615.0253 and 462.9903 were attributed to dehydrated galloyl-HHDP-hexoside and dehydrated HHDP-hexoside, respectively. They indicated the consecutive loss of two gallyol moieties from the fragment ion at *m*/*z* 785.0754. Thus, **B7** was presumed to be a compound of valoneoyl-HHDP-digalloyl-hexoside-type tannin, such as isocoriariin B [15]. The fragment ions at *m*/*z* 249.0408, 275.0200, 300.996, 169.0124, and 125.0101 were attributed to the HHDP and galloyl moieties in **B7**, which further confirmed the supposed structure.

A molecular weight of 2202 Da was assigned to **B8**, based on the doubly deprotonated ion at *m*/*z* 1100, with a retention time of 15.51 min. The fragment ion at *m*/*z* 1417.0668 resulted from the loss of valoneoyl-galloyl-hexoside moiety, such as isorugssin F. This indicated the appearance of a **B1** moiety, the dimer conjugated by HHDP-galloyl-hexoside and valoneoyl-digalloyl-hexoside, such as gemin D (**B2**) and isorugssin F. The fragment ion at *m*/*z* 633.0761 indicated the occurrence of the HHDP-galloyl-hexoside moiety, such as gemin D (**B2**), by the 1568-Da mass loss of a dimer conjugated with two valoneoyl-digalloyl-hexoside, such as isorugssin B. Based on these results, **B8** was identified as a trimer of HHDP-galloyl-hexoside and two valoneoyl-digalloyl-hexoside, such as cornusiin F [38]. Moreover, the fragment ions at *m*/*z* 450.9950, 783.0518, and 1567.1799 indicated the occurrence of the VTL moiety, dehydrated valoneoyl-digalloyl-hexoside moiety, such as isorugssin F, and the moiety resulting from the loss of an HHDP-galloyl-hexoside moiety, such as gemmin D, from the pseudo-parent ion. These findings supported the stipulated structure of **B8**.

A molecular weight of 2354 Da was assigned to **B9-1**, based on the doubly deprotonated ion at *m*/*z* 1176.0540, with a retention time of 15.51 min. The fragment ion at *m*/*z* 1417.1671 resulted from the loss of a valoneoyl-digalloyl-hexoside moiety, such as isorugssin B, indicating the appearance of **B1** moiety, a dimer conjugated by HHDP-galloyl-hexoside, and valoneoyl-digalloyl-hexoside, such as gemmin D and isorugssin F. Fragment ion at *m*/*z* 633.0614 indicated the occurrence of gemmin D by the 1720-Da mass loss of a dimer conjugated with valoneoyl-digalloyl-hexoside and valoneoyl-trigalloyl-hexoside moieties, such as isorugssin B and isorugssin F, from the pseudo-parent ion. Based on these results, **B9-1** was identified as a trimer of HHDP-galloyl-hexoside, valoneoyl-digalloyl-hexoside, and valoneoyl-trigalloyl-hexoside, such as cornusiin C [34]. Moreover, the fragment ions at *m*/*z* 450.9904, 783.052, and 935.064 indicated the occurrence of a valoneoyl moiety, dehydrated valoneoyl-digalloyl-hexoside moiety, such as isorugssin F, and dehydrated valoneoyl-trigalloyl-hexoside, such as isorugssin B. These supported the assumed structure of the **B9-1**. **B9-2**, **B9-3**, **B9-4**, and **B9-5** displayed retention times of 18.36, 17.01, 17.95, 19.62, and 22.04 min, respectively. Moreover, they exhibited similar fragment patterns as **B9-1**. Therefore, they have identified as isomers of **B9-1** with a difference in the position of the moieties linked to the hexose core or anomers with a different configuration of the anomeric hydrogen at **C-1** of the hexose core.

A molecular weight of 938 Da was assigned to **B10-1**, based on the doubly deprotonated ion at *m*/*z* 468.0396, with a retention time of 17.95 min. The fragment ion at *m*/*z* 767.05313 was attributed to the dehydrated HHDP-digalloyl-hexoside (**B4**), such as tellimagrandin I, resulting from the loss of a galloy moiety and H_2_O, which enabled the identification of **B10-1** as a HHDP-trigalloyl-hexoside-type tannin, such as tellimagrandin II [39]. The other fragment ions at *m*/*z* 614.9811 and 300.9920 indicated the occurrence of dehydrated HHDP-digalloyl-hexoside and HHDP moieties, which supported the proposed structure of **B10-1**. **B10-2**, with a retention time of 18.36 min, was also identified as a HHDP-trigalloyl-hexoside-type tannin as **B10-1** with the difference in the position of the moieties linked to the hexose core base on the similar fragment pattern.

A molecular weight of 1722 Da was assigned to **B11-1**, based on the doubly deprotonated ion at *m*/*z* 860.0783, with a retention time of 17.01 min. The fragment ion at *m*/*z* 937.0754 resulted from the loss of HHDP-digalloyl-hexoside, indicating the occurrence of valoneoyl-digalloyl-hexoside, such as isorugosin B moiety, which enabled the tentative identification of **B11-1** as a dimer of HHDP-digalloyl-hexoside and valoneoyl-digalloyl-hexoside, such as cornusiin D [39]. The other fragment ions at *m*/*z* 1419.8393 and 1087.0227 resulted from the loss of an HHDP moiety and two galloyl moieties, respectively. The fragment ion *m*/*z* of 300.9920 and 450.9815 indicated the occurrence of ellagic acid and VTL moieties, which supported the proposed structure identification. Moreover, there were four other types of tannins, such as **B11-1**, with retentions time of 17.95, 18.36, 19.62, and 21.28 min, displaying similar fragment patterns.

**B12-1** gave an [M-H]^−^ at *m*/*z* 935.0833 with a retention time of 20.29 min. It released a fragment ion at *m*/*z* 632.97, which was attributed to a HHDP-galloyl-hexoside moiety (**B2**) resulting from the loss of an HHDP (302 Da) from the pseudo-molecular ion. **B12-1** was tentatively identified as galloyl-bis-HHDP-hexoside. Moreover, the fragment ion at *m*/*z* 783.01 resulted from the loss of gallic acid from [M-H]^−^. The presence of the HHDP moiety was confirmed by the fragment ion at *m*/*z* 300.997. **B12-2**, with a retention time of 21.64 min, exhibited a galloyl-bis-HHDP-hexoside-type tannin that displayed a fragment pattern similar to that of **B12-1**. The galloyl-bis-HHDP-hexoside-type tannin has been reported in pomegranate (*Punica granatum *L.) peel [40], but that has not been detected in the seeds of *Cornus officinalis* Sieb. et Zucc.

**B13** was assigned as an ellagic acid pentoside which had a pseudo-molecular ion at *m*/*z* 433.0394 and MS/MS fragment ions at 299.997 and 300.9634; this dissociation pattern was observed in *Fragaria chiloensis* berries [41] and attributed to an ellagic acid pentoside.

**B14**, which exhibited a pseudo-molecular ion at *m*/*z* 447.0561 and fragmentation ions at *m*/*z* 315.0157 (loss of pentoside residue, 132 Da) and *m*/*z* 299.9885 (further loss of methyl) in the MS/MS spectrum, could be attributed to methyl ellagic acid pentoside. This hypothesis is in agreement with the result of the fragmentation that yielded *m*/*z* of 271 by the loss of CO_2_ from methyl ellagic acid. Methyl ellagic acid derivatives were also detected in strawberries by Seeram et al. [42].

The examples of the structures of the tannin type of B1-B14 are illustrated in Figure 7.

### 2.4. Gallotannins

**C1-1**, **C1-2,** and **C1-3**, with retention times of 4.2, 5.37, and 7.05 min, respectively, were characterized as monogalloyl-hexoside isomers. This identification was based on the [M-H]^−^ ion at *m*/*z* 331.069, and the fragment ions at *m*/*z* 169.012 indicating the loss of a hexose moiety (162 Da) and *m*/*z* 125.023 typical for the galloyl moiety resulting from the loss of the carboxylic function (44 Da) [31]. These compounds differ in the linkage position of the galloyl moiety to the hexose core.

Five compounds **C2-1** to **C2-5** (t_R_ 6.0, 7.05, 8.69, 10.09, and 10.69 min), with the same precursor ion of *m*/*z* 483.07, were identified as digalloyl-hexoside isomers, relying on the product ions at *m*/*z* 331.069, corresponding to a monogalloyl-hexoside, and resulting from the loss of galloyl moiety (152 Da) from the parent ion [31]. Moreover, the fragment ion at *m*/*z* 169.012 indicating a galloyl moiety resulted from the loss of a hexose moiety (162 Da) from the pseudo-molecular ion.

**C3-1**, **C3-2**, **C3-3**, and **C3-4** showed the same [M-H]^−^ ion at *m*/*z* 635.09. The fragment ion at *m*/*z* 465.065 corresponded to digalloyl-hexoside moiety, resulting from the loss of galloyl moiety (152 Da) and H_2_O (18 Da). Therefore, these compounds were tentatively identified as trigalloyl-hexoside isomers [43]. Additionally, the fragment ion at *m*/*z* 331.056 resulted from the consecutive loss of two galloyl moieties supporting the assignment.

**C4-1** and **C4-2**, with retention times of 18.36 and 19.62 min, respectively, both gave a pseudo-molecular ion [M-H]^−^ at *m*/*z* 787.1058, which produced the fragment ions at *m*/*z* 635.081, 465.065, and 313.0570, corresponding to the trigalloyl-hexoside, digalloyl-hexoside, and monogalloyl-hexoside moieties, respectively. These moieties resulted from the consecutive loss of three galloyl (152 Da) moieties and H_2_O (18 Da). Thus, these two compounds were tentatively assigned as tetragalloyl-hexoside isomers [43].

**C5-1**, **C5-2,** and **C5-3** produced fragment ions at *m*/*z* 787.06, indicating the presence of tetragalloyl-hexoside moiety (**C4**) in their structures. Moreover, **C5-1**, **C5-2,** and **C5-3** exhibited an [M-H]^−^ ion at *m*/*z* 939.111, which was 152 Da higher than that of **C4**, indicating the structural difference of a galloyl moiety. Then, pentagalloyl-hexoside-type tannins were assigned to **C5-1**, **C5-2,** and **C5-3** [15].

**C6-1** and **C6-2** displayed a doubly deprotonated ion at *m*/*z* 545.03, indicating a molecular weight of 1092. Fragment ion at *m*/*z* 769.05 corresponded to a dehydrated tetragalloyl-hexoside moiety, owing to the loss of two gallic acid units (152 Da) and H_2_O (18 Da) from the pseudo-molecule ion. Therefore, **C6-1** and **C6-2** were tentatively identified as hexogalloyl-hexoside isomers [31].

**C7** gave a pseudo-molecular ion [M-H]^−^ at *m*/*z* 361.0796, which liberated fragment ions at *m*/*z* 169.0129, indicating the loss of a heptose moiety (210 Da), and *m*/*z* 125.023 typical for galloyl moiety. Therefore, this compound was identified as monogalloyl-heptoside, which was in accordance with previous results [13].

**C8** was assigned as a digalloyl heptoside, which displayed an [M-H]^−^ at *m*/*z* 513.0904. It produced fragment ions at *m*/*z* 361.0781 (**C7**) and 343.071, which resulted from the loss of a gallic acid moiety (152 Da) and a further loss of water (18 Da).

The examples of the structures of the tannin type of **C1**–**C8** are illustrated in Figure 8.

### 2.5. Phenolic Acid and Their Derivatives

#### 2.5.1. Phenolic Acids

**D1** to **D6** were identified as salicylic acid, cinnamic acid, caffeic acid, coumaric acid, syringic acid, and gallic acid, respectively. Typical product ions resulted from the decarboxylation of the acidic group with *m*/*z* values of 92.9184, 102.9471, 135.0435, 119.0482, 153.0536, and 125.0229. These ions were identified in their MS/MS spectrum. Additionally, all the identified phenolic acids were characterized by comparison of their mass data with those from the reported literatures [44,45,46,47,48]. The ellagic acid was assigned to **D7**, based on the typical fragment ions m/e at 249.03, 275.02, and 300.9966. **D8**, with a precursor ion [M-H]^−^ at *m*/*z* 291.05, was assigned as brevifolin carboxylic acid, relying on the fragment ion of *m*/*z* 247.0256 resulted from the loss of carboxyl moiety. The MS data were in agreement with those previously reported for brevifolin carboxylic acid [23].

#### 2.5.2. Hydroxycinnamic Acids and Their Derivatives

**D9** and **D10** were tentatively identified as the citric acid derivatives based on the typical fragment ions of citric acid at *m*/*z* 102.9472, 146.9371, and 190.9266. **D13-1** was identified as caftaric acid (*m*/*z* 311.0364), which showed the loss of a tartaric acid moiety in the MS/MS experiment (132 Da) and a partial decarboxylation of the caffeic acid moiety resulting in fragments at *m*/*z* 179.0555 and 135.0125. This fragmentation pattern was also observed for **D13-2**, as characterized by the retention times specified in Table 1. This is presumably due to D/L isomers of tartaric acid. **D11** (*m*/*z* 295.0671) was identified as a caffeoylmalic acid, based on its fragments at *m*/*z* 133.0124, 71.0120, and 115.0020, which are characterized to malic acid moiety, as well as the typical fragment of caffeic acid at *m*/*z* 179. **D12** revealed a [M-H]^−^ ion at *m*/*z* 393.03 and a loss of 98 Da in the MS/MS, resulting in a fragment at *m*/*z* 295.0671, which, in turn, showed a fragmentation pattern identical to **D11**. Therefore it was concluded that **D12** represented a caffeoylmalic acid derivative [44,45,46,47,48].

#### 2.5.3. Hydroxybenzoic Acids and Their Derivatives

**D14** with an [M-H]^−^ ion at *m*/*z* 321.02 was assigned as digallate, based on the fragment ions at *m*/*z* 125.0230, 169.0128, as characteristic for gallic acid. **D15** revealed an [M-H]^−^ ion at *m*/*z* 333.0613 and fragments at *m*/*z* 152.9943, 109.8816 in the MS/MS, indicating a presence of a protocatechuic acid derivative. **D17** gave a deprotonated molecular ion at *m*/*z* 325.0565 and four product ions at *m*/*z* 134.0356, 149.0081, 178.0262, and 193.0488, which indicated the presence of a feruloyl moiety. The loss of 132 Da from [M-H]^−^ indicated tartaric acid substitution. Therefore, **D17** was identified as feruloyl tartaric acid. **D16** with [M-H]^−^ ion at *m*/*z* 373.1 was tentatively identified as feruloyl acid derivative, based on the observation of the typical fragment ions of feruloyl acid as described in **D17**. **D18**, **D19,** and **D20** were identified as malic acid derivatives based on the observation of the typical fragments of malic acid at *m*/*z* 71.012, 115.0021, 133.0124. Additionally, the MS/MS revealed the presence of the moieties of syringic acid (at *m*/*z* 153.0536, 182.0219, 197.0432) and gallic acid (at *m*/*z* 169.01263, 125.023) in **D18** and **D19** respectively, which enabled the tentative identification of syringoylmalic acid to **D18** with a [M-H]^−^ ion at *m*/*z* 313.05 and galloylmalic acid to **D19** with a [M-H]^−^ ion at *m*/*z* 285.02. The typical ions (at *m*/*z* 163.0381, 119.0410), characteristic for *p*-coumaric acid, were observed in the MS/MS of **D21**, **D22,** and **D23**, identified tentatively as *p*-coumaric acid derivatives. For **D21**, the product ions typical for tartaric acid (*m*/*z* 87.0067, 103.0018, 105.1979) were observed that confirmed its structure as *p*-coumaroyltartaric acid [44,45,46,47,48].

An [M-H]^−^ ion at *m*/*z* 468.9910 with a retention time of 15.51 min was observed for **D24**, producing a fragment ion at *m*/*z* 425.0161, thus indicating the loss of a carboxyl group. Additionally, typical fragments of ellagic acid at *m*/*z* 300.9971 and 299.9874 were observed. Therefore, **D24** was identified as valoneic acid bilactone isomer [3]. To our knowledge, valoneic acid bilactone has not been reported in the seeds of *Cornus officinalis* Sieb. et Zucc.

The structures of the phenolic acids are illustrated in Figure 9.

### 2.6. Non-Phenolic Compounds

Other non-phenolic compounds, such as free malic, citric, tartaric, and quinic acids, were identified. **E3-1** and **E3-2** exhibited the same fragments at *m*/*z* 71.01199, 115.00212, and 133.01247, which are characterized by the fragmentation pattern of malic acids. However, differences in the retention time were observed for **E3-1** and **E3-2** at 4.2 and 5.37 min, respectively, indicating the two isomers of malic acids. **E2-1** and **E2-2** exhibited the same [M-H]^−^ ion at *m*/*z* 149.0081, which were detected with retention times of 3.91 and 12.84 min, indicating the occurrence of two tartaric acid isomers. This identification was based on the fragments at *m*/*z* 105.0180 and 87.0066. **E4-1** and **E4-2**, exhibiting the same [M-H]^−^ ion at *m*/*z* 191.0193, were detected at 4.2 and 5.37 min, indicating the occurrence of different isomeric structures, and they were identified as quinic acids, based on the typical fragment of quinic acid at *m*/*z* 191.0193,173.0683, and 111.0068 [44,45,46,47,48].

The structures of the non-phenolic compounds are illustrated in Figure 10.

### 2.7. Total Phenolic Content (TPC)

The identified compounds indicated that the aqueous extract of the seeds was rich in tannins. We then investigated the TPC using the Folin–Ciocalteu colorimetric method, which showed a result of 79,157 ± 563 mg gallic acid equivalent (GAE)/100 g in the seed extract. Compared to the tannin-rich fruits, such as raspberries (average 233.50 mg/100 g in fresh weight), pomegranates (>10 g/100 g in dry material) peach kernels (ranging from 12.7 to 3.8 g/100 g), or the kernels of apricot cultivars (ranging from 209.4 to 10.60 mg GAE/100 g), the seeds extract of *C. officinalis* provides a new source of tannins, indicating its potential as an antioxidant for use in the food industry [49,50,51,52].

## 3. Materials and Methods

### 3.1. Solvents and Reagents

Gallic acid (GA) was obtained from the National Institutes for Food and Drug Control (Beijing, China). Acetonitrile and formic acid were of HPLC grade and purchased from Dikma Scientific (Tianjin, China). Folin–Ciocalteu reagent was obtained from Yuanyie Biotech Co., Ltd. (Shanghai, China). The water was distilled and deionized.

### 3.2. Plant Source

Mature fruits of *C. officinalis* were harvested in October 2021 from the Muzhi country in Luoyang, Henan, China. The samples were identified by Prof. Ximing Lu, Medical College, Henan University of Science and Technology, Luoyang, China. After separation from fruits, the seeds were air-dried at room temperature and then stored at 4 °C prior to analysis. Voucher specimens are maintained in the college herbarium, with certificate No. 22-7(7).

### 3.3. Sample Preparation

Owing to the structurally unstable nature of polyphenols, we performed the percolation extraction method at room temperature (20 °C). For the polyphenols, being water soluble, we used water as the extracting solvent. Percolation was performed in a stainless-steel percolator with a ball valve at the bottom. The inner diameter and height of the percolator were 5 cm and 30 cm, respectively. First, 20 mL water was poured into the percolator, then 50 g milled seeds were added. Percolation was performed at room temperature, with a flow rate of percolate 0.5 L/h using 600 mL H_2_O. Thereafter, the seed extract solvent was placed in a freeze dryer (SCIENTZ-30FG, Ningbo, China). After thermal equilibration, the shelf temperature was lowered to −40 °C and maintained for 12 h. Subsequently, the system was evacuated to a pressure of 20 Torr, and the shelf temperature was adjusted to −40 °C and held for 24 h. The shelf temperature was then raised successively to −20 °C (8 h), 0 °C (6 h), and finally, to 20 °C (2 h). The resulting amorphous samples were weighted and sealed at 4 °C for further analysis.

### 3.4. LC-MS Analysis

LC-MS analyses were carried out using a Dinonex Ultimate 3000 UHPLC system (Ultimate 3000—Thermo Scientific, Waltham, MA, USA), coupled with a quadrupole-orbitrap hybrid mass analyzer (Q-Exactive, Thermo Scientific). The chromatographic separation of the polyphenol extract was achieved on an Eclipse Plus C18 analytical column (250 mm × 4.6 m, 2.6 µm, ZORBAX, Agilent, Palo Alto, CA, USA). The column temperature was set at 30 °C. The mobile phase was composed of (A) water with 0.2% formic acid and (B) acetonitrile with 0.2% formic acid. Elution was accomplished with the following solvents gradient: 0-3 min 10% B, 18% B at 13 min and kept unchanged until 16 min, and 30% B at 25 min and kept unchanged until 30 min. Finally, the system returned to 10% B in 2 min. The flow rate and the injection volume were 0.6 mL/min and 10 μL, respectively. The acquisition was carried out in negative ionization mode (ESI-). The ESI temperature was set at 300 °C, the capillary temperature at 320 °C, and the electrospray voltage at 2.8 kV. Sheath and auxiliary gas were 30 and 5 arbitrary units, respectively. The acquisition was performed in full scan/ddMS 2 modes. The parameters were optimized as follows: (i) full scan acquisition: resolution 70,000 FWHM (at *m*/*z* 200); (ii) dd-MS 2: resolution 17,500 FWHM (at *m*/*z* 200). The normalised collision energy (NCE) was set at 30.

### 3.5. Total Phenolic Content

Diluted seeds extract (5 µL) was placed in each well of a 96-well plate and mixed with 10 µL of Folin–Ciocalteu reagent, 100 µL of H_2_O, and 50 µL of 10% sodium carbonate, and the mixture was shaken for 30 s. Total polyphenols were determined after 1 h of incubation at room temperature in the dark. The absorbance was then measured at 765 nm on a microplate reader HBS-1096A (DeTie, Nanjing, China). Gallic acid was used as a standard. The standard curve (1) with *r* as 0.9993 was prepared using different concentrations of gallic acid. The total phenolic contents were calculated as mg of gallic acid equivalent (GAE) per 100 g of the extract. The results were expressed as the mean ± standard deviations of three replications.
*Y* = 0.1227 *X* + 0.0085,(1)
where *Y* is the value of the absorbance; *X* is the concentration of samples.

## 4. Conclusions

In this study, water-soluble compounds in the seeds of *Cornus officinalis* Sieb. et Zucc. were identified using HPLC-ESI-MS/MS. A total of 97 compounds were characterized and classified as brevifolincarboxyl tannins and their derivatives, ellagitannins, gallotannins, phenolic acids and their derivatives, and non-phenolic acids. Five new types of tannins have been identified. Moreover, the method to effectively recognize the brevifolincarboxyl moiety and DHHDP moiety from the MS/MS data using typical fragment ions was summarized. Furthermore, the study of the inferred structures of tannins with technologies such as NMR and X-ray crystallography is needed in further research. The results of this study not only enrich the structures of tannin-type compounds, but also provide invaluable information for its further utilization in the industry.

## Figures and Tables

**Figure 1 molecules-28-02027-f001:**
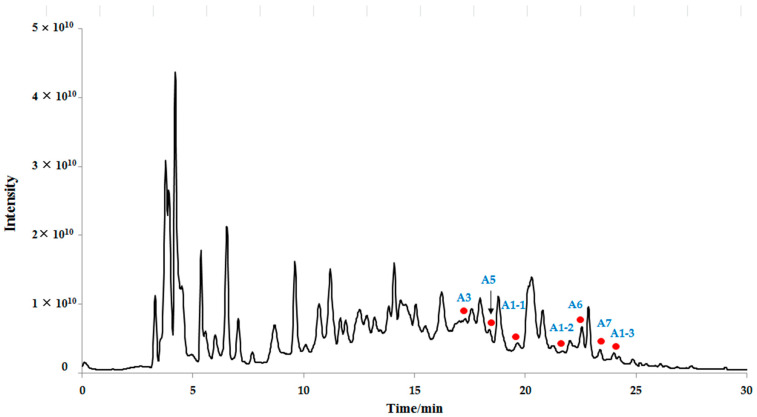
The total ion chromatogram of seeds water extract of *Cornus officinalis* Sieb. et Zucc. in the negative ESI model (the new type tannins are red dotted).

**Figure 2 molecules-28-02027-f002:**
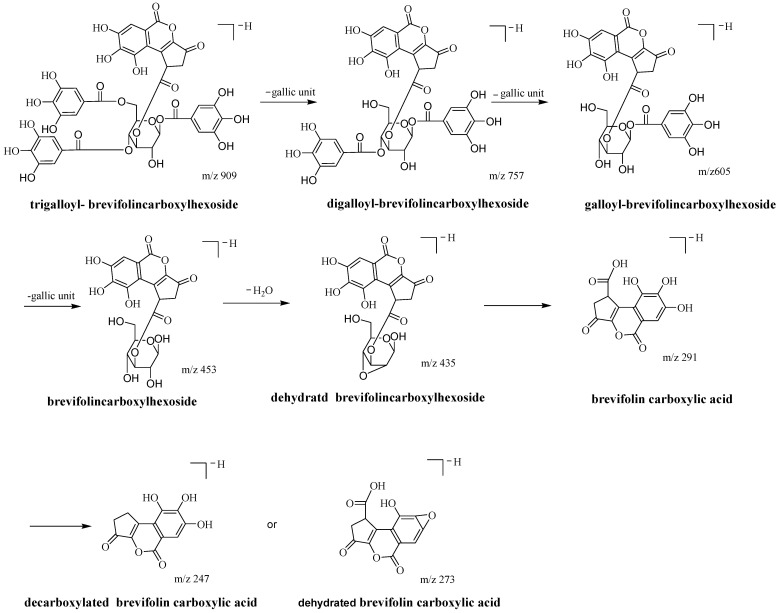
Postulated fragmentation pathways of brevifolincarboxyl-trigalloyl-hexoside (illustrated by 1,4,6-*O*-trigalloyl- 3-*O*-brevifolincarboxyl-*β*-*D*-glucose).

**Figure 3 molecules-28-02027-f003:**
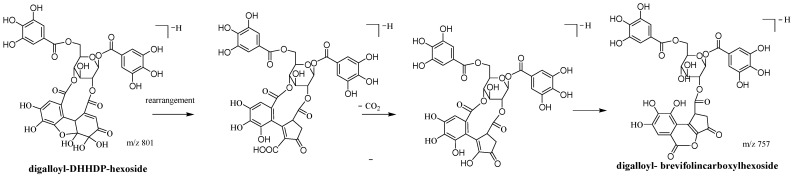
Rearrangement and decarboxylation of DHHDP moiety in digalloyl-DHHDP-hexoside (illustrated by 1,6-*O*-digalloyl-2,4-*O*-DHHDP-*β*-*D*-glucoside).

**Figure 4 molecules-28-02027-f004:**
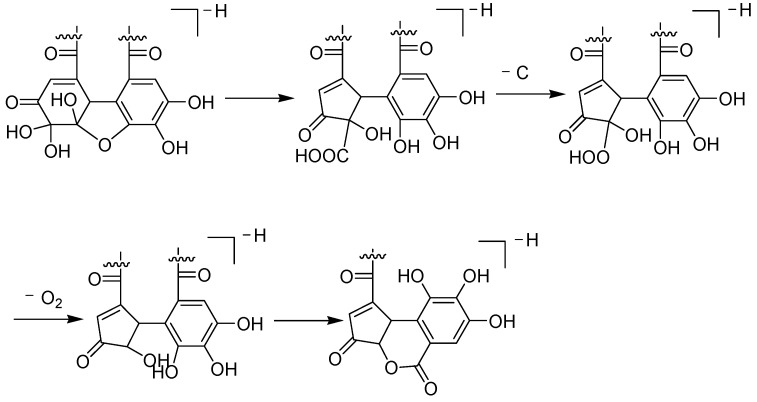
Proposed decarboxylation route of the DHHDP moiety.

**Figure 5 molecules-28-02027-f005:**
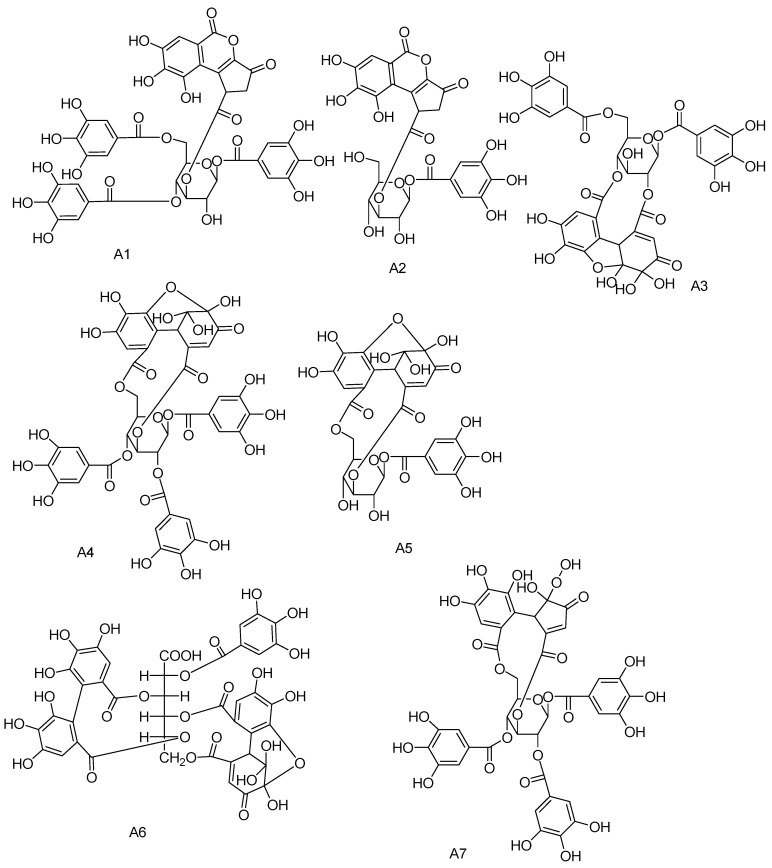
Structure of tannin type of **A1**–**A7**.

**Figure 6 molecules-28-02027-f006:**
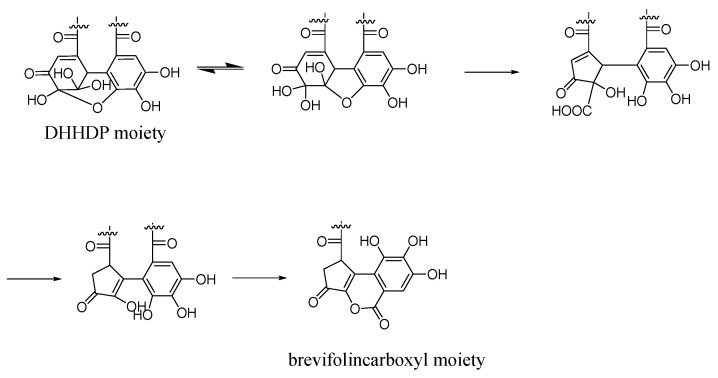
Suggested biosynthesis route of brevifolincarboxyl moiety from the DHHDP moiety.

**Figure 7 molecules-28-02027-f007:**
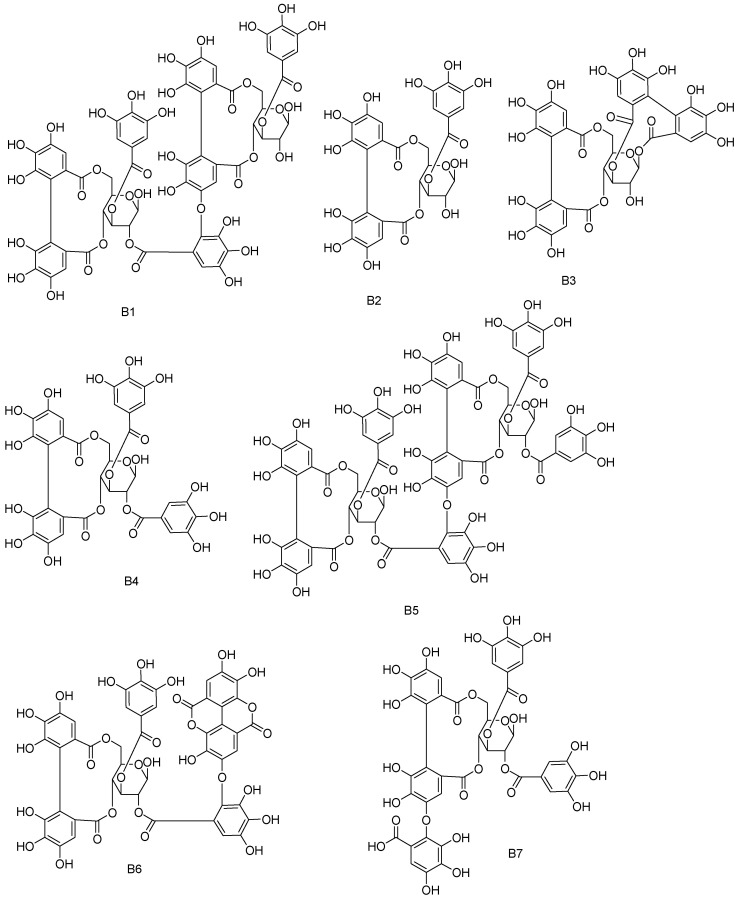
Examples of the structures of the tannin type of **B1**–**B14**.

**Figure 8 molecules-28-02027-f008:**
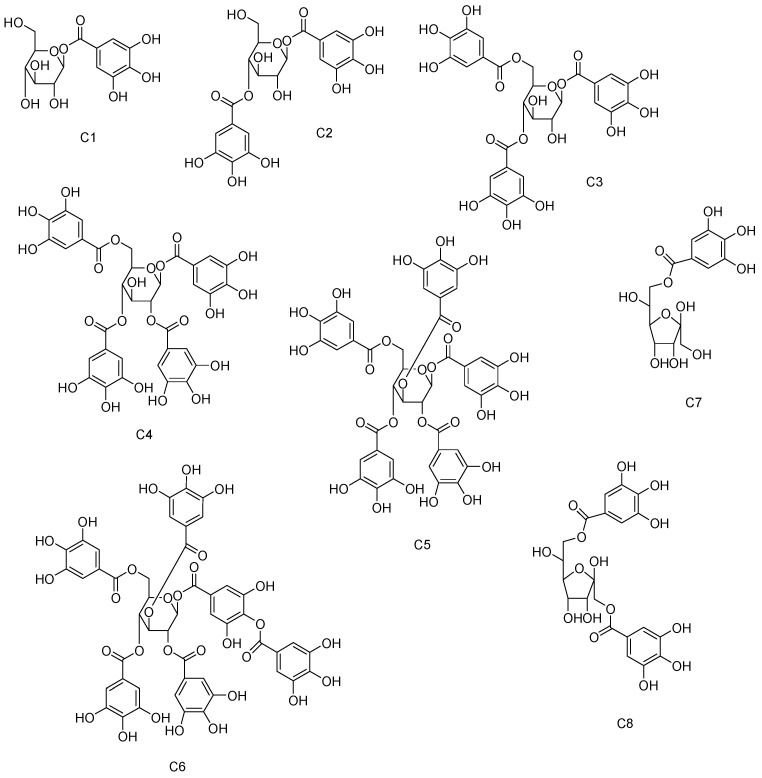
Examples of the structures of the tannin type of **C1**–**C8**.

**Figure 9 molecules-28-02027-f009:**
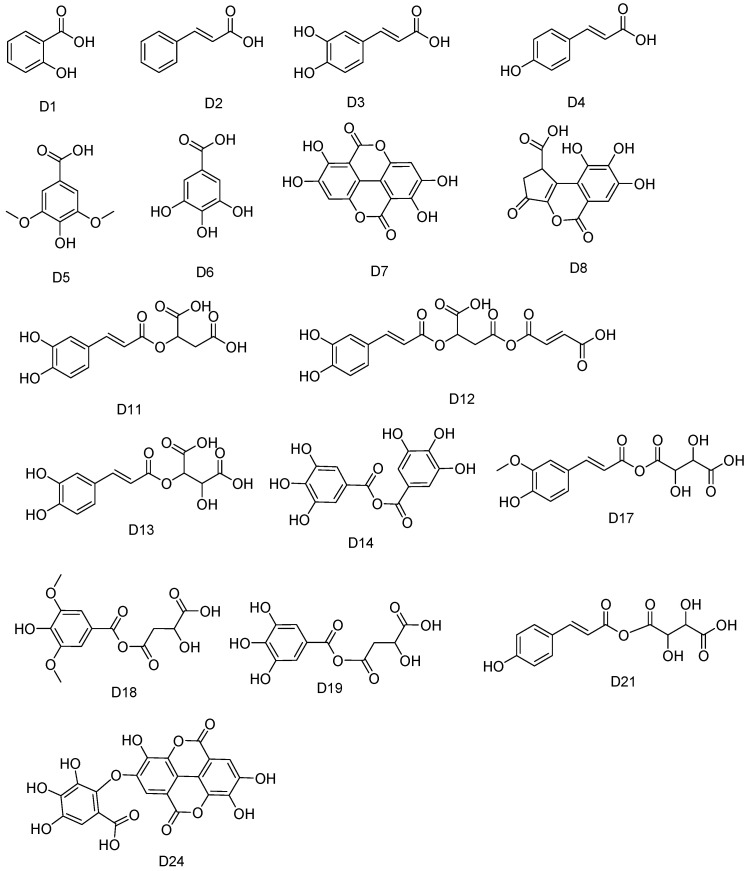
Structures of phenolic acids.

**Figure 10 molecules-28-02027-f010:**
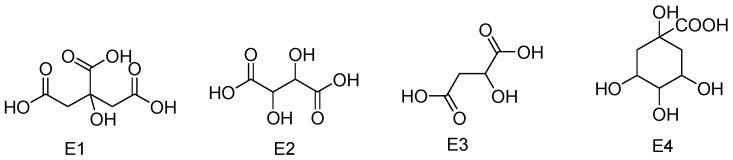
Structures of the non-phenolic compounds.

**Table 1 molecules-28-02027-t001:** Retention times (t_R_) and mass fragmentation data of compounds present in the seed water extract of *Cornus officinalis* Sieb. et Zucc.

No.	Compound		Assignment	t_R_ /min	MS /(*m*/*z*)	MS/MS Fragment Ions /(*m*/*z*)
Brevifolincarboxyl tannins and their derivatives						
1	**A1**	1	Brevifolincarboxyl-trigalloyl-hexoside	19.62	909.1014 [M-H]^−^	169.0084; 247.0257; 273.0030; 291.0139; 435.0560; 453.0302; 604.7536; 739.0454; 757.0958; 909.1014
2		2		21.28	909.0992 [M-H]^−^	169.0087; 247.0183; 273.0029; 291.0138; 435.0475; 453.057; 757.0576; 909.0732
3		3		23.80	909.1022 [M-H]^−^	169.0103; 247.014; 272.9983; 291.0465; 435.0149; 604.9794; 757.0533; 909.0967
4	**A2**		Galloyl- brevifolincarboxyl-hexoside	18.36	605.0013 [M-H]^−^	125.0230; 149.0080; 169.0128; 221.0070; 247.0255; 273.0026; 291.0116; 311.0390; 331.0200; 378.02; 383.04; 387.02; 435.03; 453.0401; 465.04; 463.04726; 587.0662; 605.0013
5	**A3**		Digalloyl-DHHDP-hexdside	17.01	801.0795 [M-H]^−^	125.0068; 169.0126; 191.03401 219.0929; 247.0251; 273.0023; 291.0128; 363.02; 374.07; 378.87; 427.02; 435.055; 445.03; 466.81; 597.02; 621.88; 627.73; 757.0929; 765.0580; 783.0775
6	**A4**		DHHDP-trigalloylhexoside	19.62	953.0915 [M-H]^−^	169.0084; 247.0257; 273.0031; 291.0140; 435.0559; 587.0396; 605.0547; 757.0971; 909.0996
7	**A5**		Monogalloyl-DHHDP-hexoside	18.36	649.1073 [M-H]^−^	125.0230; 169.0128; 465.0664; 479.0852; 497.09307; 649.1073
8	**A6**		DHHDP-HHDP-galloyl-gluconic acid	22.55	967.1068 [M-H]^−^	169.0124; 231.0274; 247.0253; 249.0377; 257.0086; 273.0068; 275.0203; 291.0096; 300.9964; 382.999; 399.01; 427.008; 445.02; 465.047; 581.05473; 597.05606; 749.05748; 765.0391; 917.0799; 935.0724
9	**A7**		Peroxide product of DHHDP-trigalloylhexoside	23.39	941.1275 [M-H]^−^	247.02572; 291.01381; 435.05605; 453.0698; 587.0662; 605.0627; 739.0645; 757.0970; 843.47; 909.10025
Ellagitannins						
10	**B1**	1	Dimer of valoneoyl-galloyl-hexoside and HHDP-galloyl-hexoside	9.6	708.0711 [M-2H]^2−^/2	169.0131; 249.0417; 275.0207; 300.9971; 450.991; 633.0780; 708.0753; 765.0594; 785.0749; 1114.88
11		2		10.09	708.0717 [M-2H]^2−^/2	169.0132; 249.0415; 275.0207; 300.9970; 450.9914; 633.0779; 708.0750; 765.0587; 783.0538; 785.0765; 936.2874; 1115.027; 1247.0553
12		3		10.69	708.0712 [M-2H]^2−^/2	125.0222; 169.0109; 249.0379; 275.0164; 300.9970; 450.9913; 633.0474; 708.0393; 765.0600; 783.0393; 785.0564; 1095.3044; 1114.9217; 1159.3596; 1247.0139; 1255.6029; 1334.6029; 1382.7864
12	**B2**	1	HHDP-monogalloyl- hexoside	10.09	633.0738 [M-H]^−^	125.02; 169.01; 249.03; 275.01; 300.99; 313.05; 331.0601; 450.98; 467.02; 481.05; 633.05
14		2		13.46	633.5754 [M-H]^−^	125.0233; 169.0112; 249.0416; 275.02081; 300.9970; 331.0615; 421.0296; 633.0784;
15	**B3**		Bis-HHDP-hexoside	10.09	783.0750 [M-H]^−^	249.0376; 275.0208; 300.9970; 450.9977; 481.0508; 765.0378; 783.0750
16	**B4**	1	Digalloyl-HHDP-hexoside	11.65	785.0776 [M-H]^−^	169.0126; 231.0303; 249.0407; 275.0201; 300.9963;419.0536; 466.0678; 483.0679; 568.5199; 743.0063; 785.0776
17		2		13.85	785.0557 [M-H]^−^	125.022; 169.01252; 249.0407; 275.0202; 300.9962; 384.7543; 483.0478; 626.5269; 785.0557
18		3		16.19	785.0776 [M-H]^−^	125.0222; 169.0130; 249.0407; 275.0201; 300.9963; 419.0283; 445.0278; 457.5856; 483.1278; 615.0389; 633.0501; 785.0776
19	**B5**	1	HHDP-digalloyl-hexoside dimer	12.51	784.0739 [M-2H]^2−^/2	125.02; 169.01324; 249.04253; 275.0206; 300.9970; 450.9906; 597.04; 613.04; 633.06; 699.04; 765.03; 784.0739; 785.0741; 935.06; 1084.85; 1266.88; 1398.82; 1479.9;
20		2		13.46	784.0769 [M-2H]^2−^/2	125.0222; 169.0111; 231.0283; 249.0380; 275.1222; 300.9921; 450.9812; 633.0624; 699.0546; 765.0389; 784.0325; 785.1371; 935.0390; 1182.3044; 1266.8373
21		3		14.38	784.0769 [M-2H]^2−^/2	125.0209; 169.0112; 231.0281; 249.0341; 275.0122; 300.9921; 450.9909; 633.0603; 699.0471; 765.0389; 784.0325; 785.07436; 935.0625; 1266.8373;
22		4		15.51	784.0769 [M-2H]^2−^/2	125.0222; 169.0116; 231.0283; 249.0380; 275.0121; 300.9871; 450.9908; 633.0643; 699.0467; 765.0390; 784.0322; 785.07436; 935.0650; 1266.8801; 1464.8974
23		5		17.95	784.0769 [M-2H]^2−^/2	125.0221; 169.0131 231.0283; 249.0378; 275.0163; 300.9920; 450.9910; 633.0602; 699.0458; 765.0387; 784.0320; 785.0743; 935.0391; 1266.8801
24		6		19.62	784.0769 [M-2H]^2−^/2	125.0221; 169.0111 231.0283; 249.0379; 275.0120; 300.9920; 450.9908; 633.0626; 699.0622; 765.0381; 784.0528; 785.0956; 935.0664; 1085.8745; 1348.183
25	**B6**		EA-(HHDP-galloyl)-galloyl-hexoside	14.38	542.0338 [M-H]^−^	300.9970; 450.9709; 542.5338; 633.0634; 765.0208; 783.0566; 785.0763
26	**B7**		Gallic acid etheric HHDP-digalloyl-hexoside	15.51	953.0909 [M-H]^−^	169.0124; 249.0408; 275.0200; 300.9963; 444.97; 462.9903; 597.0273; 615.0253; 765.0391; 783.0339; 785.0754; 909.0972; 953.0909
27	**B8**		Trimer of HHDP-galloyl-hexoside and two valoneoyl-digalloyl-hexoside	15.51	1100.0655 [M-2H]^2−^/2	169.0113;275.0207; 249.03796; 300.9969; 450.99503;613.0458; 633.0761; 765.05627;783.0518; 1015.5890; 1100.0655; 1427.1950; 1417.0668; 1567.1799
28	**B9**	1	Trimer of HHDP-galloyl-hexoside, valoneoyl-digalloyl-hexoside and valoneoyl-trigalloyl-hexoside	15.51	1176.0540 [M-2H]^2−^/2	169.0111; 231.0281; 249.0378; 275.0163; 300.9968; 450.9904; 633.0614; 765.0374; 783.0731; 935.0649; 1091.5994; 1176.0540; 1247.1002; 1417.1671; 1569.1285; 1719.1229; 2050.6591; 2103.4512
29		2		17.01	1176.0549 [M-2H]^2−^/2	169.0111; 231.0282; 249.0379; 275.0164; 300.9969; 450.9906; 633.0612; 765.0577; 783.0743; 785.0735; 935.0640; 1091.600; 1176.054; 1247.1003; 1417.117; 1567.1327; 1719.1293; 2052.0692
30		3		17.95	1176.0904 [M-2H]^2−^/2	169.0112; 231.0281; 249.0378; 275.0163; 300.9969; 450.9903; 633.0611; 765.0364; 783.0725; 785.0724; 935.0904; 1091.597; 1176.090; 1247.0928; 1417.117; 1567.1253; 1719.0564;
31		4		19.62	1176.1170 [M-2H]^2−^/2	169.0112; 231.0249; 249.0379; 275.0164; 300.9970; 450.9906; 633.6614; 765.0375; 783.0783; 785.0733; 935.0643; 1091.6337; 1176.0928; 1247.1001; 1417.1170; 1567.1280; 1719.2001; 2051.6284
32		5		22.04	1176.5870 [M-2H]^2−^/2	169.0090; 231.0282; 249.0378; 275.0164; 300.9969; 450.9906; 633.6618; 765.0379; 783.0949; 785.0731; 935.0652; 1091.5993; 1176.5870; 1247.0970; 14717.0670; 1567.1222; 1719.0538; 2052.0695; 2370.8410
33	**B10**	1	HHDP-trigalloyl-hexoside	17.95	468.0396 [M-2H]^2−^/2	125.0231; 169.0108; 249.0377; 275.0168; 300.9920; 392.0275; 468.0396; 614.9811; 767.0531
34		2		18.36	468.0395 [M-2H]^2−^/2	125.0231; 169.0128; 275.0204; 300.9969; 316.0311; 392.0363; 468.03952; 614.98; 767.0310
35	**B11**	1	Dimer of HHDP-digalloyl-hexoside and valoneoyl-digalloyl-hexoside	17.01	860.0783 [M-2H]^2−^/2	125.0222; 169.0111; 231.0280; 249.0378; 275.0164; 300.992; 450.9815; 633.0482; 699.0474; 765.0383; 784.0733; 860.0783; 937.0754; 1087.0227; 1419.839
36		2		17.95	860.0278 [M-2H]^2−^/2	169.0132; 249.0418; 275.0207; 300.99693; 450.99036; 597.05; 765.05; 775.07; 785.03; 860.0278; 935.06; 937.10097; 937.10; 1087.0970; 1249.10; 1419.1430
37		3		18.36	860.0780 [M-2H]^2−^/2	125.0208; 169.0111; 231.0280; 249.0378; 275.0164; 300.9919; 450.9906; 633.0596; 699.0244; 765.0384; 784.0520; 860.0780; 937.1008; 1087.098; 1267.0512
38		4		19.62	860.0545 [M-2H]^2−^/2	125.0228; 169.0112; 231.0249; 249.0379; 275.0164; 300.992; 450.9911; 633.0595; 699.0457; 765.0577; 784.0731; 860.0545; 937.0749; 1087.098; 1419.194
39		5		21.28	860.0549 [M-2H]^2−^/2	125.0220; 169.0110; 231.0246; 249.0378; 275.0163; 300.992; 450.9906; 633.0466; 699.0358; 765.0386; 784.0525; 860.0549; 937.0749; 1087.063; 1093.38; 1139.656
40	**B12**	1	Galloyl-bis-HHDP-hexoside	20.29	935.0833 [M-H]^−^	139.01322; 300.997; 581.0539; 597.05; 632.97; 749.05547; 783.01; 917.0757; 935.08338
41		2		21.64	935.0808 [M-H]^−^	169.0110; 231.028; 247.0223; 275.0165; 300.9720; 597.0398; 749.0552; 783.0803; 917.0495; 935.0660
42	**B13**		Ellagic acid pentoside	17.58	433.1349 [M-H]^−^	299.9979; 300.9634; 387.1276; 433.0394; 450.8096?
43	**B14**		Methyl ellagic acid pentoside	22.55	447.0570 [M-H]^−^	270.9903; 298.9806; 299.9885; 314.0053; 315.0157; 332.0889; 333.090; 447.05616
Gallotannins						
44	**C1**	1	Mono-galloyl- hexoside	4.2	331.0672 [M-1]^−^	125.0230; 169.0128; 211.0240; 331.0694
45		2		5.37	331.0694 [M-1]^−^	125.0230; 169.0128; 211.0239; 331.0694
46		3		7.05	311.0639 [M-1]^−^	125.0231; 169.01289; 311.0639
47	**C2**	1	Di-galloyl- hexoside	6.0	483.0780 [M-H]^−^	125.0230; 169.0128; 311.0690; 483.0778
48		2		7.05	483.0781 [M-H]^−^	125.0230; 169.0128; 311.0690; 483.0778
49		3		8.69	483.0779 [M-H]^−^	125.0230;169.01281; 193.0132; 211.0239; 271.0454; 313.0572;
50		4		10.09	483.1459 [M-H]^−^	125.0231; 169.0128; 193.0130; 211.0239; 241.0342; 271.0451; 313.0572; 331.06935; 483.1459
51		5		10.69	483.0776 [M-H]^−^	125.0230; 169.0127; 211.0139; 271.0451; 313.05721; 483.0776
52	**C3**	1	Trigalloyl- hexoside	11.9	635.0928 [M-H]^−^	125.0229; 169.0124; 313.0568; 465.0655; 635.0928
53		2		13.46	635.0924 [M-H]^−^	125.0231; 169.0128; 211.0239; 271.0454; 313.0569; 465.0657; 483.07735; 635.0924
54		3		14.39	635.0923 [M-H]^−^	125.0229; 169.0124; 313.0568; 465.0655; 635.0928
55		4		15.05	635.0930 [M-H]^−^	125.0232; 169.0129; 211.024; 271.0452; 313.0574; 465.0648; 483.0775; 635.0930
56	**C4**	1	Tetragalloyl- hexoside	18.36	787.1058 [M-H]^−^	125.0229; 169.0125; 313.0570; 465.0657; 635.09187; 787.1058
57		2		19.62	787.1058 [M-H]^−^	125.0229; 169.0125; 313.0567; 465.0653; 635.09194; 787.1058
58	**C5**	1	Penta-galloyl-hexoside	20.77	939.1118 [M-H]^−^	125.02282; 169.01253;313.05; 403.05; 447.04; 465.04; 513.06?; 617.05; 635.05; 787.06; 939.1118
59		2		21.28	939.1122 [M-H]^−^	125.0228; 169.0124; 313.0570; 465.0658; 787.10365; 939.1122
60		3		22.04	939.1120 [M-H]^−^	125.0228; 169.0124; 313.0570; 465.0658; 617.0827; 787.1036; 939.1120
61	**C6**	1	Hexa-galloyl- hexoside	23.39	545.03 [M-2H]^2−^/2	125.0230; 169.0128; 241.0350; 317.0350; 393.0477; 431.0635; 447.0562; 465.0650; 545.0487; 601.0745; 617.0484; 769.0929
62		2		24.00	545.03 [M-2H]^2−^/2	125.0231; 169.0219; 241.0345; 317.0403; 393.0465; 431.0635; 447.0562; 465.0662; 545.0591; 601.0754.; 617.0731; 769.0745; 896.8940; 935.1472
63	**C7**		Monogalloyl-heptoside	5.37	361.0796 [M-H]^−^	125.0232; 169.0129; 211.0244; 241.0348; 271.04527; 361.0796
64	**C8**		Digalloyl-heptoside	10.69	513.0904 [M-H]^−^	125.0230; 169.0128; 211.0240; 271.0451; 343.0711; 361.0781; 513.0904
Phenolic acids and their derivatives						
65	**D1**		Salicylic acid	0.1	136.86152 [M-1]^−^	92.91849; 136.86152
66	**D2**		Cinnamic acid	0.1	146.93 [M-1]^−^	58.85718; 87.92379; 102.94717
67	**D3**	1	Caffeic acid	12.84	179.03318 [M-1]^−^	135.04356; 179.03318
68		2	Caffeic acid isomer	15.51	179.03323 [M-1]^−^	135.0436; 179.03323
69	**D4**		*p*-Coumaric acid	16.19	163.03804 [M-1]^−^	119.04825; 163.03804
70	**D5**		Syringic acid	17.01	197.04325 [M-1]^−^	121.0280; 153.0536; 182.0217; 197.04325
71	**D6**		Gallic acid	6.52	169.01284 [M-1]^−^	125.0229; 169.01284
72	**D7**		Ellagic acid	21.28	300.99706 [M-1]^−^	300.99706
73	**D8**		Brevifolin carboxylic acid	13.19	291.01377 [M-1]^−^	173.022;191.0317; 203; 219; 247.02568; 291.01377
74	**D9**		Citric acid derivative 1	0.1	280.8698 [M-1]^−^	102.9471; 118.9424; 130.8875; 146.9368; 162.8384; 174.8779; 190.9262; 218.8685; 236.8784; 262.85823; 280.8698
75	**D10**		Citric acid derivative 2	0.1	336.85 [M-1]^−^	102.9472; 146.9371; 190.92668
79	**D11**		Caffeoylmalic acid	3.91	295.06706 [M-1]^−^	71.01203; 115.0020; 133.0124; 179.05567; 295.06706
77	**D12**		Caffeoylmaloyl Fumaric acid	3.91	393.03 [M-1]^−^	71.01206; 79.9552; 96.95802; 115.0021; 135.0125; 179.0555; 295.06659
78	**D13**	1	Caftaric acid	12.51	311.03643 [M-1]^−^	87.00687; 135.0421; 149.0081; 179.03326; 311.03643
79		2		12.84	311.04 [M-1]^−^	59.01217; 87.00672; 135.04366; 149.00816; 179.0333
80	**D14**		Digallic acid or digallate	12.51	321.02 [M-1]^−^	125.02308; 169.01285
81	**D15**		Protocatechuic acid derivative	3.58	333.06135 [M-1]^−^	78.95685; 109.8816; 152.9943; 171.004; 241.0099; 333.06135
82	**D16**		Feruloyl acid derivative	12.51	373.113 [M-1]^−^	193.04875; 149.05987; 373.113
83	**D17**		Feruloyl tartaric acid	17.58	325.05 [M-1]^−^	59.01226; 87.00679; 103.0017; 134.03563; 149.0081;178.0262; 193.0488;
84	**D18**		Syringoylmalic acid	17.01	313.05 [M-1]^−^	71.0121; 115.0022; 121.02804; 133.0125;153.0536; 182.0219;197.0432
85	**D19**		Galloyl malic acid	8.69	285.02 [M-1]^−^	71.012; 115.00211; 133.01243; 169.01263; 125.023
86	**D20**		Malic acid derivative	15.51	505.15889 [M-1]^−^	71.0120; 101.02559; 115.0022; 127.0385; 133.0125; 227.0914; 389.1437; 487.1433; 505.15889
87	**D21**		*p*-Coumaroyltartaric acid	16.19	295.00 [M-1]^−^	59.01225; 87.0067; 103.00181; 105.1979;119.04835; 130.9971;163.03806;
88	**D22**		*p*-Coumaroyl acid derivative 1	22.55	331.08094 [M-1]^−^	109.0258; 119.0470; 163.0381; 207.0282; 287.0933; 272.0602; 287.0152; 331.08094
89	**D23**		*p*-Coumaroyl acid derivative 2	22.04	361.0927 [M-1]^−^	119.0483;139.0377;163.0381; 207.0282; 287.0471; 302.0470; 317.0244; 361.09278
90	**D24**		Valoneic acid bilactone	15.51	468.9910 [M-1]^−^	125.0271; 169.0132; 270.9862; 298.9814; 299.9814; 300.9971; 425.01618; 468.9910
Non-phenolic acids						
91	**E1**		Citric Acid	0.1	190.9274 [M-1]^−^	58.9567; 102.94717; 146.93691;190.9274
92	**E2**	1	Tartaric acid	3.91	149.00 [M-1]^−^	59.01211; 72.99147; 87.00666; 103.00173; 105.018; 149.00818
93		2		12.84	149.00 [M-1]^−^	59.01226; 72.9916; 87.00679; 103.00188; 105.016; 149.00811
94	**E3**	1	Malic acid	4.2	133.01247 [M-1]^−^	71.01199; 115.00212; 133.0124
95		2		5.37	133.01253 [M-1]^−^	71.0121; 115.00218; 133.01253
96	**E4**	1	Quinic acid	4.2	191.01939 [M-1]^−^	85.02796; 111.0068; 117.0181; 129.0174; 154.9972; 173.0683; 191.0193
97		2		5.27		85.02792; 111.0068; 191.01934

## Data Availability

The data are available upon request to the authors.

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
