# Peer review of "New Type of Tannins Identified from the Seeds of Cornus officinalis Sieb. et Zucc. by HPLC-ESI-MS/MS"

_molecules, 2023, doi:10.3390/molecules28052027_

Round 1

Reviewer 1 Report

Reviewer’s comments

The authors have identified new 5 types of tannins, as polyphenol compounds, in the seeds of Cornus officinalis using HPLC-ESI-MS/MS. The experiments are well-designed and the results are clearly presented. This study will enrich the structure database and open new horizons in further research and application of identified compounds, especially in the industry. The manuscript should be checked by Grammarly. Also, the references should be corrected in accordance with the Instructions for Authors. Thus, I recommend the publication of the manuscript after Minor Revision.

The recommended corrections have been noticed in the pdf file of the manuscript.

Reviewer 2 Report

The article is very well designed and well written and clearly presents the results of the research.

It is only necessary to fill in empty spaces (blanks) in some sentences throughout the text. My suggestions are noted in the article itself.

Reviewer 3 Report

This manuscript investigated the phenolic profile of Cornus officinalis Sieb. Et Zucc. seeds water extract through HPLC-ESI-MS/MS. 90 polyphenols in total were identified, five new types of which were reported for the first time. Overall, the structure identification process was elaborated in detail and rigorously. The findings of this study were meaningful for other researchers focusing on the structure identification of phenolics and the further valorization of Cornus officinalis Sieb. Et Zucc. seeds. However, before being accepted, there are some questions that the authors need to figure out:

Line 138-140: are there any other reports supporting the ions at m/z 247, 273, and 291 as the typical indicator ions for the brevifolincarboxyl moiety identification?

In Figure 2, if the authors claimed m/z 273 as one of the typical indicators of brevifolincarboxyl, why is this fragment present not in the postulated fragmentation pathway in Figure 2?

Line 497: Are there any discussions and references for the identification of E1 as citric acid?

Line 510-511: The aqueous extract of seeds was rich in tannins, but TPC was used to reflect the total phenolic content, not just tannins because flavonoids were also claimed as one of the major phenolics (line 42). Is it possible to determine the content of tannins? Like using colorimetric or HPLC method?

Line 529: Is it common to store the seeds at -4°C?

Line 568-575 (Section 3.5): Please provide the reference and standard curve.

Can the authors provide examples of structures of gallotannins (C1-C8), phenolic acids and derivatives (D1-D24), and other non-phenolic compounds (E1-E4) like Figure 5 and Figure 7?

Reviewer 4 Report

The manuscript is interesting and well presented. The scientific quality and originality of this work is usfficient for publication in Molecules. Conclusions are adequately supported by data. The results obtained during the experiments were described and interpreted in detail. The main part of the manuscript is the description of the results of separation and fragmentation of compounds dissolved in aqueous seed extract. In my opinion the manuscript requires confirmacion of the proposed structures (confirming example).

Reviewer 5 Report

The authors claim that the extract reacts with the FeCl3 solution, indicating the presence of polyphenols, but the method by which this is done is not explaine;  I recommend adding the part in the paragrap. 
